# A Short Corticosteroid Course Reduces Symptoms and Immunological Alterations Underlying Long-COVID

**DOI:** 10.3390/biomedicines9111540

**Published:** 2021-10-26

**Authors:** Alberto Utrero-Rico, María Ruiz-Ruigómez, Rocío Laguna-Goya, Estíbaliz Arrieta-Ortubay, Marta Chivite-Lacaba, Cecilia González-Cuadrado, Antonio Lalueza, Patricia Almendro-Vazquez, Antonio Serrano, José María Aguado, Carlos Lumbreras, Estela Paz-Artal

**Affiliations:** 1Instituto de Investigación Sanitaria 12 de Octubre (imas12), 28041 Madrid, Spain; rryruiz@gmail.com (M.R.-R.); rociolagunagoya@gmail.com (R.L.-G.); earrietao@gmail.com (E.A.-O.); marta.chivite@gmail.com (M.C.-L.); ceciliagcuadrado@gmail.com (C.G.-C.); lalueza@hotmail.com (A.L.); patricia.almendro.vazquez@gmail.com (P.A.-V.); aserranoh@gmail.com (A.S.); jaguadog1@gmail.com (J.M.A.); carlos.lumbreras@salud.madrid.org (C.L.); estela.paz@salud.madrid.org (E.P.-A.); 2Department of Internal Medicine, Hospital Universitario 12 de Octubre, 28041 Madrid, Spain; 3Department of Immunology, Hospital Universitario 12 de Octubre, 28041 Madrid, Spain; 4Department of Medicine, School of Medicine, Universidad Complutense de Madrid, 28041 Madrid, Spain; 5Unit of Infectious Diseases, Hospital Universitario 12 de Octubre, 28041 Madrid, Spain; 6Department of Immunology, Ophthalmology and ENT, School of Medicine, Universidad Complutense de Madrid, 28041 Madrid, Spain

**Keywords:** long-COVID, immunological alterations, corticosteroids

## Abstract

Despite the growing number of patients with persistent symptoms after acute SARS-CoV-2 infection, the pathophysiology underlying long-COVID is not yet well characterized, and there is no established therapy. We performed a deep immune profiling in nine patients with persistent symptoms (PSP), before and after a 4-day prednisone course, and five post-COVID-19 patients without persistent symptoms (NSP). PSP showed a perturbed distribution of circulating mononuclear cell populations. Symptoms in PSP were accompanied by a pro-inflammatory phenotype characterized by increased conventional dendritic cells and augmented expression of antigen presentation, co-stimulation, migration, and activation markers in monocytes. The adaptive immunity compartment in PSP showed a Th1-predominance, decreased naïve and regulatory T cells, and augmentation of the PD-1 exhaustion marker. These immune alterations reverted after the corticosteroid treatment and were maintained during the 4-month follow-up, and their normalization correlated with clinical amelioration. The current work highlights an immunopathogenic basis together with a possible role for steroids in the treatment for long-COVID.

## 1. Introduction

SARS-CoV-2 has spread globally and caused the coronavirus disease 2019 (COVID-19) pandemic, which continues to be a current challenge. Moreover, there is growing evidence of recovered patients who show persistent symptoms months after acute infection, which include fatigue, dyspnea, ageusia, arthralgia, myalgia, and anosmia, amongst more than 50 described different symptoms [1,2,3,4,5,6,7,8]. Similar persistent symptoms have also been observed after other viral infections [9], including Severe Acute Respiratory Syndrome (SARS) and Middle East Respiratory Syndrome (MERS) coronavirus [10].

The persistence of symptoms beyond the acute phase of SARS-CoV-2 infection has been termed long-COVID, and it has been estimated to affect 30–80% of COVID-19 patients [1,5,11,12]. Despite its high incidence, long-COVID is still an unknown condition. While there are studies showing an increased risk of persistent symptoms in more severe patients [1], other studies report that the development and severity of persistent symptoms do not seem to be associated with the severity of the acute infection [4,13]. So far, there are no clear laboratory or radiological features to diagnose long-COVID, and the condition is thought to be heterogeneous, with potentially several phenotypes with different presentations and prognosis [14,15]. Virus-induced changes, endothelial dysfunction, immunological alterations, and/or expected sequelae have been proposed as responsible for long-COVID [16]; however, the underlying pathophysiology of these long-term effects has not yet been well-characterized. In addition, there are no established protocols for the management of these patients. In this study, we set out to broaden the understanding of the pathophysiology underlying long-COVID and to test the effectiveness of corticosteroids in this condition, with the aim of helping define diagnostic and treatment guidelines to manage these patients with COVID-19 sequelae.

## 2. Materials and Methods

### 2.1. Study Participants

On April 2021, 14 patients who had had COVID-19 confirmed by positive RT-PCR between 3 and 13 months ago were recruited in Hospital 12 de Octubre (Madrid, Spain). Nine out 14 described persistent symptoms (PSP), while five described a completely healthy status (NSP) (see Appendix A). Eight PSP patients received a 4-day course of corticosteroids (prednisone, 30 mg/day), after which another sample was collected (TTP). A total of 16 mL of blood was collected in K2EDTA tubes for immune phenotyping. Clinical symptoms were evaluated before and after treatment. A second post-treatment sample was obtained in six patients with persistent symptoms (TTP+4M) 4 months after the course of corticosteroids. The institutional Clinical Research Ethics Committee approved the study protocol (reference no. 20/167), and all patients signed an informed consent.

### 2.2. Flow Cytometry

Peripheral blood mononuclear cells (PBMC) were obtained after Ficoll–Paque centrifugation at 2000 rpm for 30 min. Cells were counted with viability. A total of 1 × 106 PBMC were stained with anti-CD3-V450 (BD Bioscience, San Jose, CA, USA), anti-CD19-V450 (BD Bioscience, San Jose, CA, USA), anti-CD56-V450 (BD Bioscience, San Jose, CA, USA), anti-CD14-FITC (BD Bioscience, San Jose, CA, USA), anti-CD33-PE-Cy7 (eBioscience, San Diego, CA, USA), anti-HLA-DR-APC (BD Bioscience, San Jose, CA, USA), anti-CD16-V500 (BD Bioscience, San Jose, CA, USA), anti-CCR2-PE (BD Bioscience, San Jose, CA, USA), anti-CCR5-APC-Cy7 (BD Bioscience, San Jose, CA, USA), and anti-CD86-PerCP-Cy5.5 (BD Bioscience, San Jose, CA, USA). Classical monocytes were defined as CD14+CD16−, intermediate monocytes as CD14+CD16+, and non-classical monocytes as CD14−CD16+. Expression of CCR2, CCR5, HLA-DR, and CD86 was evaluated and analyzed by mean intensity fluorescence (MFI). A total of 1.5 × 106 PBMC were stained with anti-CD4-KO (Beckman Coulter, Brea, CA, USA), anti-CD25-FITC (BD Bioscience, San Jose, CA, USA), and anti-CD127-PE-Cy7 (BD Bioscience, San Jose, CA, USA). Regulatory T cells (Treg) were identified as CD4+CD25+CD127low. A total of 2 × 106 PBMC were stained in a sequential staining with anti-CD3-BV510 (Biolegend, San Diego, CA, USA), anti-CD19-Spark NIR 685 (Biolegend, San Diego, CA, USA), anti-CD14-Spark Blue 550 (Biolegend, San Diego, CA, USA), anti-CD16-BUV469 (BD Bioscience, San Jose, CA, USA), anti-CD8-BUV805 (BD Bioscience, San Jose, CA, USA), anti-CD56-BUV737 (BD Bioscience, San Jose, CA, USA), anti-CD45RA-BUV395 (BD Bioscience, San Jose, CA, USA), anti-CD27-APC-H7 (BD Bioscience, San Jose, CA, USA), anti-IgM-BV570 (Biolegend, San Diego, CA, USA), anti-IgD-BV480 (BD Bioscience, San Jose, CA, USA), anti-CD141-BB515 (BD Bioscience, San Jose, CA, USA), anti-CD1c-Alexa Fluor 647, anti-CD2-PerCP-Cy5.5 (Biolegend, San Diego, CA, USA), anti-CD28-BV650 (Biolegend, San Diego, CA, USA), anti-CD279-BV785 (Biolegend, San Diego, CA, USA), anti-CD39-BUV661 (BD Bioscience, San Jose, CA, USA), anti-CD127-APC-R700 (BD Bioscience, San Jose, CA, USA), anti-IgG-BV605 (BD Bioscience, San Jose, CA, USA), anti-CXCR5-BV750 (BD Bioscience, San Jose, CA, USA), anti-CCR5-BUV563 (BD Bioscience, San Jose, CA, USA), anti-CXCR3-PE-Cy7 (Biolegend, San Diego, CA, USA), anti-CCR6-BV711 (Biolegend, San Diego, CA, USA), anti-CCR7-BV421 (Biolegend, San Diego, CA, USA), anti-CD38-APC/Fire810 (Biolegend, San Diego, CA, USA), anti-CD314-BUV615 (BD Bioscience, San Jose, CA, USA), anti-CD337-PE/Dazzle594 (Biolegend, San Diego, CA, USA), anti-CD159a-APC (Miltenyi Biotec, Bergisch Gladbach, Germany), anti-CD159c-PE (Miltenyi Biotec, Bergisch Gladbach, Germany), anti-CD95-PE-Cy5 (BD Bioscience, San Jose, CA, USA), anti-CD57-FITC (Biolegend, San Diego, CA, USA), anti-CD4-cFlour-YG584 (Cytek Biosciences, Fremont, CA, USA), anti-CD123-Super Bright 436 (Invitrogen, Waltham, MA, USA), anti-TCRγδ-PerCP-eFluor710, anti-CD11c-eFluor450 (Invitrogen, Waltham, MA, USA), anti-HLA-DR-PE/Fire810 (Biolegend, San Diego, CA, USA), and anti-CD20-Pacific Orange (Life Technologies, ThermoFisher Scientific, Waltham, MA. USA), as previously published [16]. A total of 34 different immune populations were identified as shown in Appendix A. Cells were acquired using a Cytek^®^ Aurora flow cytometer (Cytek Biosciences, Fremont, CA, USA) and analyzed by FlowJo V10 (Tree Star, Ashland, OR, USA). T-distributed stochastic neighbor embedding (tSNE) graphs were performed by using dowsample and tSNE FlowJo V10 plugins. Markers used for each tSNE are specified in the figure legends.

### 2.3. Monocytes Stimulation and Cytokine Detection

For ex vivo stimulation and cytokine production, monocytes were isolated from PBMC (EasySep Human monocytes enrichment kit w/o CD16, StemCell, Vancouver, BC, Canada), according to the manufacturer’s instructions. The purified monocyte fraction was stained with anti-CD14-FITC (BD Bioscience, San Jose, CA, USA), and purity was evaluated via flow cytometry. Purity of monocytes was >90% in all cases. The purified monocyte fraction was plated in a 96-well flat bottom plate (100,000 monocytes per well) and incubated for 1 h in RPMI1640 supplemented with 10% fetal bovine serum (FBS), 1% penicillin-streptomycin, 1% L-glutamine, and 1% sodium-piruvate at 37 °C. After that, culture media were removed, and cells were washed with warmed PBS. New warmed culture media were added with 10 ng/mL of LPS, and monocytes were incubated overnight. Supernatant was then collected and stored at −20 °C.

Frozen supernatant samples were thawed and centrifuged. Eleven cytokines were detected using the MILLIPLEX Map Human Cytokine/Growth Factor panel (Merck KGaA, Darmstadt, Germany): IL6, TNFa, IL10, IL1b, GM-CSF, IL18, IL8, MIP1a, IL1Ra, CCL2, and CXCL10, according to the manufacturer’s instructions. Plates were analyzed by a Luminex 100 platform (One lambda, ThermoFisher Scientific, Waltham, MA. USA).

### 2.4. Lymphocyte Stimulation and Intracellular Staining

CD14- cells isolated from PBMC with CD14 Microbeads (Miltenyi Biotec, Bergisch Gladbach, Germany) were incubated with Ionomycin (Ion, 1.25 ng/mL), Phorbol Myristate Acetate (PMA, 40 ng/mL), and Brefeldin A (BFA, 5 µg/mL) for 4 h. Following activation, CD14- cells were stained with anti-CD4-FITC (BD Bioscience, San Jose, CA, USA) and anti-CD8-PerCP (BD Bioscience, San Jose, CA, USA) for 30 min, then fixed and permeabilized using Inside Stain Kit (Miltenyi Biotec, Bergisch Gladbach, Germany), followed by intracellular staining with IL2-PE (Miltenyi Biotec, Bergisch Gladbach, Germany), IFNγ-APC (Miltenyi Biotec, Bergisch Gladbach, Germany), and TNFα-APC (Miltenyi Biotec, Bergisch Gladbach, Germany). Acquisition was performed using a five-laser Cytek^®^ Aurora flow cytometer (Cytek Biosciences, Fremont, CA, USA) and analyzed by FlowJo V10 (Tree Star).

### 2.5. Auto-Antibodies Detection

Frozen plasma samples were used for autoantibody detection in our clinical immunology laboratory. IgA-aB2GPI antibodies were quantified by an enzyme-linked immunosorbent assay using QUANTA Lite B2 GPI IgA (INOVA Diagnostics Inc., San Diego, CA, USA). Cut-off was established at 20 U/mL with the 99th percentile of the healthy normal population. IgG-aB2GPI, IgM-aB2GPI, IgG-aCL, and IgM-aCL antibodies were determined by BioPLex 2200 Antiphospholipid Syndrome (APLS) antibodies kits (Bio-Rad, Hercules, CA, USA). The cut-off to consider all four antibodies positive was 18 UI/mL, according to the 99th percentile evaluated in a healthy population. The vasculitis antineuthropil cytoplasmic autoantibodies (ANCA), i.e., anti-myeloperoxidase (MPO) and anti-proteinase 3 (PR3), were evaluated by an immunoCAP assay (Thermo-Fisher, Phadia AB, Uppsala, Sweden). Cut-off for both ANCA was 0.2 AI. The following antinuclear autoantibodies (ANA) were analyzed using BioPlex 2200 ANA screen (Bio-Rad Laboratories, Hercules, CA, USA) run on a BioPlex 2200 System: dsDNA, chromatin, ribosomal P, Ro52 (SS-A), Ro60 (SS-B), La (SS-B), Sm, Sm-RNP complex, RNP-A, RNP-68, Scl-70, centromere B, and Jo-1. The cut-off for all ANA detected by BioPlex was 1.0 AI, except for anti-RNP-A, anti-RNP-68, and anti-dsDNA antibodies, which were 2.0 AI, 2.0 AI, and 20 AI, respectively, as recommended by the manufacturer. Both myositis and neuronal autoantibodies were determined by the immunodot technique (AutoPlexDot from D-tek and PNS9 DIVER BLOT IgG from Ravo Diagnostika GmbH, respectively). The myositis autoantibodies profile included PL-7, PL-12, EJ, SRP, Mi-2, MDA-5, TIF1-y, Ku, and PM-Scl 100. The neuronal autoantibodies profile included GAD65, SOX1, Ma2, Ma1, amphiphysin, CV 2 (CRMPA5), Ri, Yo, and HuD.

### 2.6. SARS-CoV-2 Specific Response by Fluorospot

PBMC were seeded in duplicate at 300,000 cells per well in IFN-γ IL-2 FluoroSpotTM plates (MabTech, Nacka Strand, Sweden) with cell culture medium containing RPMI, 1% L-glutamine, 1% penicillin/streptomycin, 10% fetal bovine serum, and anti-CD28 mAb (1 µg/mL). Cells were supplemented with 15-mer overlapping peptides covering the S1 domain of the S glycoprotein (166 peptides) (SARS-CoV-2 S1 scanning pool, MabTech, Nacka Strand, Sweden), the nucleoprotein (N protein) (102 peptides) (Epitope Mapping Peptide Set (EMPS) SARS-CoV-2 NCAP-1, JPT), and the M protein (53 peptides) (EMPS SARS-CoV-2 VME1, JPT) at a final concentration of 1 µg/mL. Negative control wells lacked peptides, and positive control wells included anti-CD3 mAb (MabTech, Nacka Strand, Sweden). Assays were incubated for 16–18 h at 37 °C. Spots were counted using an automated IRISTM FluoroSpot Reader System (MabTech, Nacka Strand, Sweden). To quantify antigen-specific responses, spots of the negative control wells were subtracted from the mean spots test wells, and the results were expressed as IFN-γ or IL-2 producing spot forming units (SFU) per 10^6^ PBMCs. Results were excluded if negative control wells had >80  IFN-γ sfu/10^6^ PBMCs or positive control wells had <400 IFN-γ sfu/106 PBMCs. Reponses were considered positive if the results were at least three times higher than the mean of the negative control wells and above of the following antigen-specific cut-off values: > 25IFN-γ sfu/10^6^ PBMCs for the S1 domain of the S glycoprotein, >14  IFN-γ sfu/10^6^ PBMCs for the N protein, and >21  IFN-γ sfu/10^6^ PBMCs for the M protein.

### 2.7. Statistics

Continuous numeric variables have been represented as mean and standard error of the mean (SEM). NSP and PSP group variables were compared by using a Mann–Whitney U test, while paired PSP and TTP group variables were compared by using a Wilcoxon signed rank test. Categorical variables have been represented as N and percentage. Principal component analysis (PCA) of scaled variables and biplot was performed using factoMineR and factorextra R packages. Hierarchical clustering analysis and heatmap representation were performed with log10-transformed and center-scaled variables by using the ComplexHeatmap R package. In the figures, grey regions depict percent populations ranges, if available, obtained from groups of representative healthy donors in our clinical immunology laboratory.

All the analysis was performed in R v 4.0.3 and GraphPad Prism 8 (San Diego, CA, USA). Differences were considered statistically significant when *p*-value < 0.05.

## 3. Results and Discussion

### 3.1. Study Patients

We recruited nine patients who had suffered mild acute COVID-19 (WHO ordinal scale of 1–3) [17], on average 9 months ago (range 3 to 13 months), and maintained persistent symptoms (persistent symptoms patients, PSP) (Figure 1a). Eight out of nine (88.9%) patients were female with a median age of 42 (range from 30 to 61). They described the following symptoms: arthralgia (7/9, 78%), myalgia (8/9, 89%), dyspnea (8/9, 89%), asthenia (8/9, 89%), paresthesia (7/9, 78%), dizziness (5/9, 56%), anxiety (5/9, 56%), insomnia (4/9, 44%), headache (5/9, 56%), brain fog (4/9, 44%), anosmia (3/9, 33%), and dysgeusia (2/9, 22%). Eight PSP received a 4-day corticosteroid course of oral prednisone, 30 mg/24 h. A first blood sample was obtained just before the beginning of the treatment, and a second blood sample was obtained after finishing it, i.e., 8 h after the last prednisone dose (Treated patients, TTP). After completion of the treatment, all patients described complete or partial recovery of their symptoms. P6 showed complete recovery from all symptoms and was therefore discharged from the clinic. Overall, the symptoms that improved the most after treatment were arthralgia (6/7 TTP, 86%), myalgia (6/8 TTP, 75%) and asthenia (4/7 TTP, 57%). Detailed data of each patient can be found in Appendix A.

### 3.2. Long-COVID Patients show Immunological Alterations Involving Innate and Adaptive Immune Populations

We aimed to investigate the existence of immune alterations that might underlie long-COVID. A deep immune profiling was performed before (PSP) and after treatment (TTP). It was compared with that obtained from a control group including five hospital workers (60% females, median age of 44, range from 25 to 62) fully recovered from mild acute COVID-19 disease (WHO ordinal scale of 1–3) between 3 and 13 months ago and without any ongoing symptoms (No symptoms patients, NSP) (Figure 1a).

Principal component analysis with 46 immune features, which included 34 immune cell populations analyzed by spectral flow cytometry and expression of lymphocyte and monocyte markers, showed differences among PSP, TTP, and NSP, which allowed an accurate identification and separation of the three patient groups (Figure 1b). The immune variables that differed the most between NSP and PSP belonged to Dimension 1 (Figure 1c). In this dimension, TTP appeared closer to NSP than to PSP. According to the immune populations detected by flow cytometry, the unsupervised hierarchical clustering analysis correctly classified all the three groups: PSP and NSP grouped distantly from each other, while TTP occupied an intermediate position and showed more similarity with NSP than with themselves before treatment (Figure 1d). These results suggested that symptomatic patients present immune alterations that are corrected to levels similar to asymptomatic patients after corticosteroid treatment.

### 3.3. Altered Dendritic Cells and Monocytes Are Found in Long-COVID and Are Corrected with Corticosteroid Treatment

We examined the innate immune system for alterations potentially related to persistent symptoms. PSP had significantly increased conventional CD11c+ dendritic cells (cDC; *p* < 0.01) and natural killer T (NKT)-like cells (*p* < 0.05), and significantly decreased CD123+ plasmacytoid DC (pDC; *p* < 0.05) and total circulating innate lymphoid cells (ILC; *p* < 0.001), when compared to NSP (Figure 2a). After corticosteroid treatment, NKT lymphocytes and, more markedly, cDC dropped, and ILC frequency increased, to levels similar to those of NSP. We did not find any differences in natural killer (NK) cell subpopulations, except for decreased terminal NK cells in PSP compared with NSP (*p* < 0.05) that recovered after treatment (*p* < 0.01, Appendix A). Within the monocyte compartment, there was a trend towards overrepresentation of intermediate monocytes (CD14+CD16+) in PSP, as observed during acute COVID-19 [18,19,20], while classical (CD14+CD16−) and non-classical (CD14−CD16+) monocytes showed similar levels to NSP (Appendix A). The surface expression of CCR2, CCR5, HLA-DR, and CD86 in the non-inflammatory, classical monocytes was similar between NSP and PSP (Appendix A). However, intermediate and, in particular, inflammatory, non-classical monocytes in PSP showed a significantly higher expression of these chemotactic and co-stimulatory markers than in NSP (Appendix A). After corticosteroid treatment, a decreased expression of all these markers on the surface of non-classical monocyte was observed (Appendix A). Upon in vitro stimulation with LPS, isolated CD14+ monocytes from PSP showed a trend towards a higher capacity to secrete the pro-inflammatory cytokines GM-CSF, IL-18, IL-1b, IL-6, MIP-1a, and TNFa than NSP monocytes, which reached significance in the case of IL-8 (Appendix A). The production of cytokines significantly dropped after the corticosteroid treatment. Overall, we observed a pro-inflammatory phenotype in PSP, even over a year after the acute SARS-CoV-2 infection, characterized by an increase in antigenic presentation capacity and augmented expression of markers of migration and activation in pro-inflammatory monocytes. In vitro stimulated monocytes from PSP showed increased potential to secrete cytokines. This inflammatory profile was reverted by the short course of corticosteroids.

### 3.4. Profound Alterations in Subset Distribution and Activation and Exhaustion Markers in T Cells from Long-COVID Patients which Revert after Corticosteroid Treatment

We next analyzed the different compartments of the adaptive immunity. We observed a striking decrease in naïve CD4+ T cells in PSP compared with NSP (*p* < 0.001, Figure 2b) and an increase in memory CD4+ T cells (*p* < 0.01). Circulatory follicular helper T cells (cTfh) were also reduced in PSP (*p* < 0.05). The CD4+ T cell compartment in PSP was not only skewed towards memory but it was also significantly enriched in activated (HLA-DR+CD38+) (*p* < 0.001) and pro-apoptotic, exhausted (CD95+PD1+) cells (*p* < 0.001, Figure 2c). Subpopulations of memory CD4+ T cells and cTfh were also identified according to previous reports [21,22,23]. In both, a decrease of Th2 and Th17 cells towards a substantial increase of Th1 cells (*p* < 0.05 and *p* < 0.001, respectively) marked a dysregulated compartment in PSP in comparison to NSP (Figure 2d,e). Finally, in agreement with the inflammatory environment in PSP, regulatory T cells (Treg) were decreased in these patients (*p* < 0.001, Figure 2f). Similar alterations were found in the CD8+ T cell compartment of PSP. There was a decrease in naïve CD8+ T cells in PSP compared with NSP (*p* < 0.05), together with a nonsignificant increase of TEMRA CD8+ T cells (Figure 2g). Activated and senescent cells were also significantly enriched within CD8+ T cells (*p* < 0.01 and *p* < 0.001, respectively) (Appendix A). All of these imbalances in the CD4+ and CD8+ T cell compartment normalized after corticosteroid treatment (Figure 2b–g) and were not related to persistent alterations from acute infection in the complete blood count (e.g., lymphopenia or neutrophilia) as both NSP and PSP had normal hemogram values (Appendix A). The scarcity of naïve and overrepresentation of memory T cells, together with the high expression of activation markers in PSP, could suggest a continuous antigenic stimulation. All NSP and PSP included in the study had a current negative antigen test and RT-PCR for SARS-CoV-2 from nasopharyngeal samples; however, this does not exclude the possibility of viral persistence in other locations such as central nervous or digestive systems [24,25,26]. Alterations in the T cell compartment have been described in convalescent COVID-19 patients, particularly the persistence of activated CD8+ T cells for up to 6 months post-infection [19,27]. Despite the aforementioned disturbances in the distribution of T cell populations, the secretion of interleukin (IL)-2, interferon gamma (IFNg), and tumor necrosis factor alpha (TNFa) by CD4+ or CD8+ T cells after polyclonal in vitro stimulation was similar between PSP and NSP, and corticosteroid treatment did not have a clear effect on cytokine synthesis (Appendix A). The lack of increased cytokine production by activated T cells in PSP, together with their elevated expression of PD-1, raises the possibility of these memory CD4+ and CD8+ T cells being exhausted. The use of PMA and ionomycin instead of more physiological stimuli such as CD3, CD28 antibodies, or viral peptide pools may be considered a limitation in the assessment of cytokine production. Of note, there was an imbalance towards Th1 cells in PSP, which coincides with the findings by Shuwa et al. [27] in convalescent patients at shorter time post-infection and indicates the existence of an inflammatory state in these patients, which improves upon corticosteroid treatment. Alterations in the naïve and memory B cell balance were also observed in PSP. These patients with long-COVID had a lower level of naïve B cells and a higher level of non-class-switched memory B cells, while class-switched memory B cell and plasmablast levels were similar to those in NSP (Figure 2h). This phenotypic imbalance was reverted after the prednisone treatment. There were no differences in IgM, IgG, or IgA levels between NSP and PSP (Appendix A). Long-COVID has been proposed to be associated with the presence of autoimmune antibodies [28,29]. In order to elucidate if autoantibodies may associate with the persistence of symptoms in PSP, we analyzed a wide group of antibodies including antiphospholipid (aPL), antineutrophil cytoplasmic (ANCA), antinuclear (ANA), and myositis and neuronal-related autoantibodies. The great majority of autoantibodies were negative in all patients (Appendix A), suggesting that autoimmunity did not play a role in the persistent symptoms in these patients. Despite the alterations found in T and B lymphocyte subpopulations, all PSP vaccinated against SARS-CoV-2 developed anti-Spike specific cellular and humoral responses.

### 3.5. The Corticosteroid-Induced Correction of Immunological Alterations in Long-COVID Persists four Months after Treatment

We evaluated if corticosteroid-corrected immunological alterations observed in PSP are maintained over time. For this purpose, another blood sample was obtained and analyzed 4 months after corticosteroid treatment, and symptoms were concurrently evaluated. None of the six PSP followed up received additional treatment during this time. Four out of six patients (66.7%) referred to at least a relapse of arthralgia or myalgia, but in all cases, those symptoms were less intense than the first time and improved progressively without any treatment.

In comparison to NSP, PSP showed a significantly lower level of CD4 lymphocytes four months after treatment; however, naïve, memory, cTfh, and Treg lymphocytes were similar to those found in NSP (Figure 2b,f). Similar results were found when we analyzed Th1, Th2, and Th17 memory and cTfh subpopulations, all of which showed similar distributions in PSP 4 months after treatment to in NSP (Figure 2d,e). Normalization of all of these populations was accompanied by maintained low levels of pro-apoptotic, exhausted (CD95+PD1+) cells (Figure 2c).

In the CD8 T lymphocyte compartment, there was a tendency towards increased CD8 T lymphocytes in PSP 4 months after treatment in comparison to NSP. There were no differences in the level of pro-apoptotic, exhausted (CD95+PD1+) CD8 T cells between PSP and NSP (Appendix A), and naïve, TEMRA, and TCM CD8 subpopulations showed similar distributions in PSP to those found in NSP (Figure 2g). Overall, these results showed that the profound alteration found in the T lymphocyte compartment of PSP was reverted by the corticosteroid course, and the corrected immunophenotype was maintained at least during the 4 month follow-up.

The present study has several limitations. The reduced number of patients in each cohort and the selection of PSP who reported particularly severe myalgia and/or arthralgia, together with other symptoms, limit the conclusions of our study and do not allow the characterization of different long-COVID phenotypes. The duration of the clinical amelioration beyond 4 months post-treatment was not investigated, and whether a longer corticosteroid treatment could be more appropriate is still unclear. In addition, further studies evaluating the persistence of SARS-CoV-2, or viral parts of it, in locations other than nasopharynx in post-COVID-19 patients will be of interest.

## 4. Conclusions

We provide evidence that the persistent symptoms of long-COVID may be accompanied by immunological alterations characterized by inflammation and reduced immune regulation. These alterations were observed over a year after acute infection and included an imbalance towards Th1 predominance, with increased inflammatory markers and low level of Treg. A short course of corticosteroids reverted these immune alterations and led to clinical amelioration of persistent symptoms, both of which were maintained 4 months after treatment. The current study provides an immunopathogenic basis for long-COVID and suggests that corticosteroids may be effective in this situation.

## Figures and Tables

**Figure 1 biomedicines-09-01540-f001:**
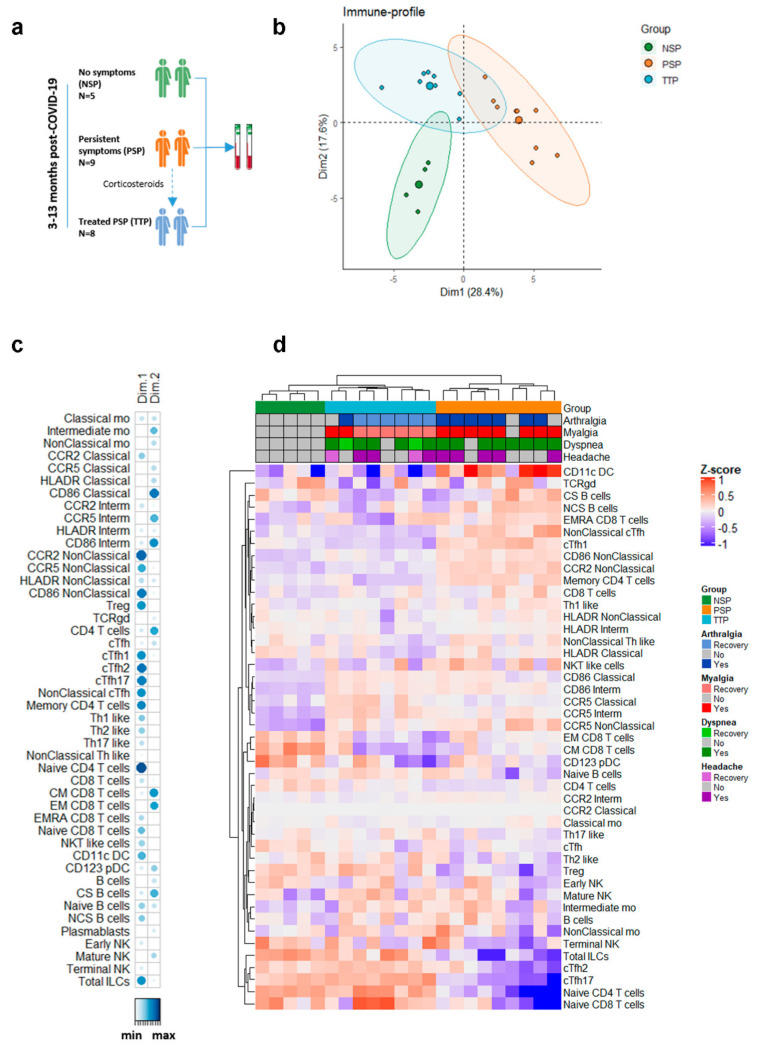
Deep immune profiling detected alterations in post-COVID-19 patients with persistent symptoms, which were corrected with a short course of corticosteroids. (**a**) Study design included three groups of post-COVID-19 patients: five subjects who recovered completely (green, NSP) and nine subjects with persistent symptoms (orange, PSP), eight of whom were sampled again after 4 days of prednisone treatment (blue, TTP). (**b**) Principal component analysis (PCA) based on all flow cytometry data. (**c**) Contribution of each variable to Dimensions 1 and 2 of PCA. (**d**) Unsupervised hierarchical clustering analysis and heatmap of log10-transformed and center-scaled variables.

**Figure 2 biomedicines-09-01540-f002:**
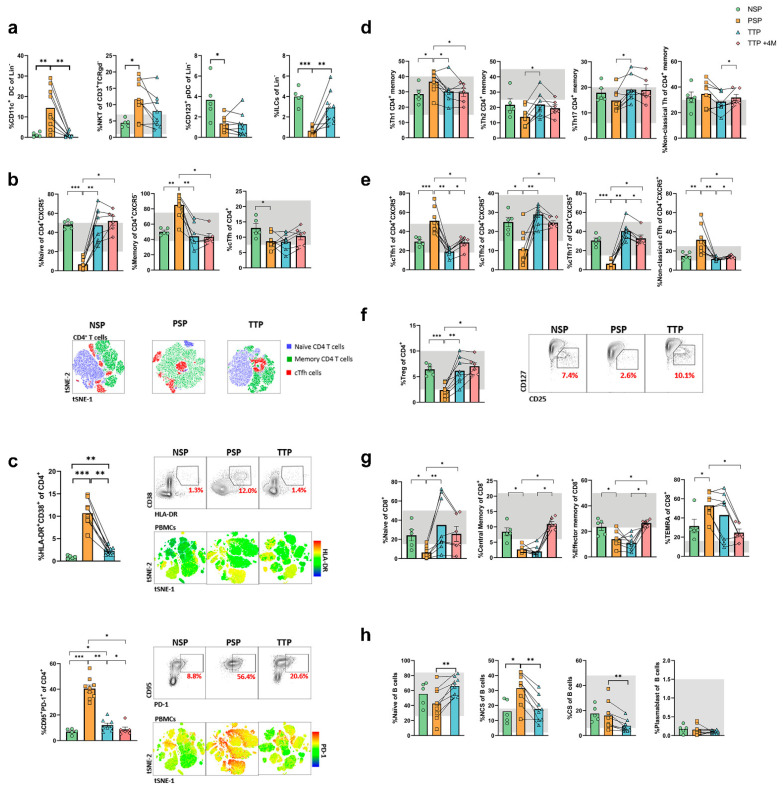
Immunological alterations in PSP include dysregulated innate and adaptive populations together with higher expression of activation factors. Comparison between NSP, PSP, TTP, and TTP+4M of (**a**) conventional dendritic cells (CD11b+ DC), natural killer T lymphocytes (NKT), plasmacytoid DC (CD123+ pDC), and innate lymphoid cells (ILC); (**b**) naïve, memory, and follicular helper T CD4+ lymphocytes; (**c**) expression of HLA-DR and CD38 and expression of CD95 and PD-1 on CD4+ T lymphocytes; (**d**) subsets of memory CD4+ T cells based on CXCR3 and CCR6 expression; (**e**) subsets of follicular helper T CD4+ lymphocytes; (**f**) regulatory T cells (Treg); (**g**) naïve, central memory (TCM), effector memory (TEM), and effector memory re-expressing CD45RA (TEMRA) CD8+ T cells; and (**h**) naïve, non-class-switched memory (NCS), class-switched memory (CS), and plasmablast B cells. Grey regions depict percent populations ranges, if available, obtained from groups of representative healthy donors in our clinical immunology laboratory. In (**b**), tSNEs were performed by using all 37 fluorochromes (see Methods); and in (**c**), tSNEs were performed in total CD4+ cells and using CXCR5, CD27, CCR7, CD45RA, CCR6, and CXCR3. *, *p* < 0.05; **, *p* < 0.01; ***, *p* < 0.001.

## Data Availability

Data and code supporting the findings of this study are available from the corresponding author on request.

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
