# Peer review of "A Short Corticosteroid Course Reduces Symptoms and Immunological Alterations Underlying Long-COVID"

_biomedicines, 2021, doi:10.3390/biomedicines9111540_

Round 1
Reviewer 1 Report
To whom it may concern,
The study is well managed, and the results are timely with high significant therapeutic outcomes for the PSP patients (reduction of long-COVID-associated syndroms among others).
However, before publication, I would suggest few minor corrections and verifications:
- minor: I would replace ameliorates in the title by decreases. The terms ameliorates symptoms seem to be a little strange.
- major: I noticed a significant differences for the proportion men versus women in the two compared groups. Indeed, the PSP group has around 90% of women and the NSP only the half of the patients. I would recommend the authors to add fewer patients to have similar proportion and to ensure that all data reflect the differences between PSP and NSP, and not between 90 and 50% of women.
- minor 2: In the methodology section (page 3, line 98), the authors write that they used flow cytometry to evaluate the purity of collected monocyte. However, no indication is given regarding the % of purity per se.
- major 2; for the experiment 2.4 (lymphocyte stimulation), the authors used PMA and ionomycin to activate their CD14-depleted cells. Since cell activation with PMA and ionomycin is absolutely not physiologic, I would recommend the authors to redo the assay using either polyclonal stimulation (with anti-Cd3 and anti-CD28 antibodies) or antiviral stimulation (with CEF peptides).
Author Response
Biomedicines-1424959
Answers to Reviewers Comments
Reviewer(s)' Comments to the Authors:
Reviewer: 1
To whom it may concern,
The study is well managed, and the results are timely with high significant therapeutic outcomes for the PSP patients (reduction of long-COVID-associated symptoms among others).
However, before publication, I would suggest few minor corrections and verifications:
Question 1:
Minor: I would replace ameliorates in the title by decreases. The terms ameliorates symptoms seem to be a little strange.
Answer:
Following the reviewer’s suggestion, we have changed the term “ameliorates” for “reduces” in the title (page 1). The new title reads:
“A short corticosteroid course reduces symptoms and immunological alterations underlying long-COVID”
Question 2:
Major: I noticed a significant differences for the proportion men versus women in the two compared groups. Indeed, the PSP group has around 90% of women and the NSP only the half of the patients. I would recommend the authors to add fewer patients to have similar proportion and to ensure that all data reflect the differences between PSP and NSP, and not between 90 and 50% of women.
Answer:
When we recruited the patients for the NSP control group we matched them in age and sex with the PSP. In PSP there were 8 women out of 9 patients, while in NSP there were 3 women out of 5 patients, that is, 88.9% and 60% respectively. This information appears in the manuscript in the supplementary table S1 and in the text:
Results, section 3.1 (page 4): “Eight out of 9 (88.9%) patients were female with a median age of 42 (range from 30 to 61).”
Results, section 3.2 (page 6): “It was compared with that obtained from a control group including 5 hospital workers (60% females, median age of 44, range from 25 to 62) fully recovered from mild acute COVID-19 disease (WHO ordinal scale of 1-3) between 3 and 13 months ago and without any ongoing symptoms (No symptoms patients, NSP) (Figure 1a).”
Despite the number of men and women being different in each group, the proportion of them is similar amongst groups as shown by the Fisher’s exact test:
|
|
NSP |
PSP |
|
Male |
2 (40%) |
1 (11%) |
|
Female |
3 (60%) |
8 (89%) |
When proportions were compared using Fisher´s exact test, the p-value was 0.51.
Nevertheless, the number of patients in this study is low and this remains as a limitation. In future studies, we will recruit a higher number of patients and maintain the groups matched for gender and other relevant variables.
Question 3:
Minor 2: In the methodology section (page 3, line 98), the authors write that they used flow cytometry to evaluate the purity of collected monocyte. However, no indication is given regarding the % of purity per se.
Answer:
The purity of isolated monocytes was above 90% in all cases. This is an example of a purification:
As suggested by the reviewer, we have added the information regarding purity in the Materials and Methods, section 2.3 (page 3), as follows:
“Purified monocyte fraction was stained with anti-CD14-FITC (BD), and purity was evaluated via flow cytometry. Purity of monocytes was >90% in all cases. Purified monocyte fraction was plated…”.
Question 4:
Major 2; for the experiment 2.4 (lymphocyte stimulation), the authors used PMA and ionomycin to activate their CD14-depleted cells. Since cell activation with PMA and ionomycin is absolutely not physiologic, I would recommend the authors to redo the assay using either polyclonal stimulation (with anti-Cd3 and anti-CD28 antibodies) or antiviral stimulation (with CEF peptides).
Answer:
Despite being widely used, we agree that PMA and ionomycin stimulation is non-physiological. Thus, other type of stimuli may have been more appropriate to evaluate differences in the capacity of T lymphocytes to secrete pro-inflammatory cytokines between NSP and PSP. Unfortunately, we performed all experiments on fresh samples and we do not have frozen cell samples available from these patients’ cohorts to evaluate cytokine production using a different stimulus. Nevertheless, we will take this suggestion into consideration when planning future experiments and we have now mentioned this issue as a limitation in Results, section 3.4 (page 9) as follows:
“…The use of PMA and ionomycin instead of more physiological stimuli such as CD3, CD28 antibodies or viral peptide pools may be considered a limitation in the assessment of cytokine production. Of note,…”

Reviewer 2 Report
In this study Utrero-Rico A. and colleagues characterize the immune cell compartment in individuals who fully recovered from COVID-19 and those who present post-acute-COVID-19 symptoms. Authors find the immune disbalance in post-acute-COVID-19 patients which is characterized by increased pro-inflammatory state. Short treatment with corticosteroids corrected for the immune disbalance and reduced the symptoms. Given the severity of COVID-19 pandemic the study if of high interest and importance. The major problem is the lack of placebo control which reduces the significance of the findings. Apart from that the study is well performed and the manuscript is well written. I have minor comments:
- In the Introduction (lines 38-40) authors provide with the estimation of 30-80% of COVID patients developing long-COVID. Authors should be more precise here and clarify that these estimates relate to hospitalized COVID-19 cases. Furthermore, authors should cite here other important studies, for example: doi:10.1016/S0140-6736(20)32656-8; doi:10.1001/jama.2020.12603; doi:10.1016/j.cmi.2020.09.052; doi:1093/qjmed/hcaa178; doi:10.1016/j.jinf.2020.08.029; doi:10.1002/jmv.26368; doi:10.7326/M20-5661 etc., especially that none of the referred literature reports the estimation of 80%.
- In figure 2d authors separate CD4+ T cells into different subpopulations (Th1, Th2 and Th17) based on CCR6 and CXCR3 expression. They should provide with the reference for such a strategy. Furthermore, in phrase in lines 267-270 authors mention 17 cytokines. Which cytokines? This sentence is not clear. Authors should clarify it.
- Authors should discuss the discrepancy between the results in figure 2d and e and supplementary figure 4. While figure 2 shows increase in Th1 cells in PSP, figure S4 shows no significant changes between patients in this cell population.
- Figures 2 and 3 are partially redundant. Authors should not show the same data in two different figures. This needs to be reorganized.
Author Response
Reviewer: 2
In this study Utrero-Rico A. and colleagues characterize the immune cell compartment in individuals who fully recovered from COVID-19 and those who present post-acute-COVID-19 symptoms. Authors find the immune disbalance in post-acute-COVID-19 patients which is characterized by increased pro-inflammatory state. Short treatment with corticosteroids corrected for the immune disbalance and reduced the symptoms. Given the severity of COVID-19 pandemic the study if of high interest and importance. The major problem is the lack of placebo control which reduces the significance of the findings. Apart from that the study is well performed and the manuscript is well written. I have minor comments:
Question 1:
In the Introduction (lines 38-40) authors provide with the estimation of 30-80% of COVID patients developing long-COVID. Authors should be more precise here and clarify that these estimates relate to hospitalized COVID-19 cases. Furthermore, authors should cite here other important studies, for example: doi:10.1016/S0140-6736(20)32656-8; doi:10.1001/jama.2020.12603; doi:10.1016/j.cmi.2020.09.052; doi:1093/qjmed/hcaa178; doi:10.1016/j.jinf.2020.08.029; doi:10.1002/jmv.26368; doi:10.7326/M20-5661 etc., especially that none of the referred literature reports the estimation of 80%.
Answer:
The prevalence of long-COVID is undoubtedly high, however, the estimates of patients developing this condition vary widely from article to article and for that reason we reported a range from 30 to 80%. For example, the paper doi:10.1001/jama.2020.12603 mentioned by the reviewer, by Carfi et al (new reference 5) reported that 87% of discharged patients had persistent symptoms. In addition, there is controversy as to whether the probability of persistent symptoms increases with disease severity or not.
We have extended the Introduction (pages 1-2) to clarify these issues and to include the studies suggested by the reviewer which have enriched the text. The second paragraph of the Introduction is now as follows:
“SARS-CoV-2 has spread globally and caused the coronavirus disease 2019 (COVID-19) pandemic which continues to be a current challenge. Moreover, there is growing evidence of recovered patients who show persistent symptoms months after acute infection, which include fatigue, dyspnea, ageusia, arthralgia, myalgia and anosmia, amongst more than 50 described different symptoms [1-8]. Similar persistent symptoms have also been observed after other viral infections [9], including Severe Acute Respiratory Syndrome (SARS) and Middle East Respiratory Syndrome (MERS) coronavirus [10].”
“The persistence of symptoms beyond the acute phase of SARS-CoV-2 infection has been termed long-COVID and it has been estimated to affect 30-80% of COVID-19 patients [1,5,11,12]. Despite its high incidence, long-COVID is still an unknown condition. While there are studies showing an increased risk of persistent symptoms in more severe patients [1], other studies report that the development and severity of persistent symptoms do not seem to be associated with the severity of the acute infection [4, 13].”
Question 2:
In figure 2d authors separate CD4+ T cells into different subpopulations (Th1, Th2 and Th17) based on CCR6 and CXCR3 expression. They should provide with the reference for such a strategy. Furthermore, in phrase in lines 267-270 authors mention 17 cytokines. Which cytokines? This sentence is not clear. Authors should clarify it.
Answer:
The expression of chemokine receptors is commonly used to define human CD4+ T cell subsets. The expression of CXCR3 is preferentially maintained by Th1 cells, CCR6 is expressed by Th17 cells and Th2 cells lack expression of both chemokine receptors (new references 21 and 23). A similar strategy for cTfh1, cTfh2 and cTfh17 has been previously described (new reference 22).
As suggested by the reviewer, we have provided these references in the manuscript (page 9, Results section 3.4):
“Subpopulations of memory CD4+ T cells and cTfh were also identified according to previous reports [21-23].”
In the same section (page 9, section 3.4), we had referred to Th2 and Th17 cells as “type 2 and 17 cytokines secreting cells”.
We realize that this wording may be confusing and have rewritten it more clearly:
The sentence: “In both, a decrease of type 2 and 17 cytokines secreting cells towards a substantial increase of type 1 (p<0.05 and p<0.001, respectively) marked a dysregulated compartment in PSP in comparison to NSP (Figure 2d,e).”
Has been changed for: “In both, a decrease of Th2 and Th17 cells towards a substantial increase of Th1 cells (p<0.05 and p<0.001, respectively) marked a dysregulated compartment in PSP in comparison to NSP (Figure 2d,e).”
Question 3:
Authors should discuss the discrepancy between the results in figure 2d and e and supplementary figure 4. While figure 2 shows increase in Th1 cells in PSP, figure S4 shows no significant changes between patients in this cell population.
Answer:
As pointed out by the reviewer, there is a discrepancy between figure 2d and e and supplementary figure 4. This apparent mismatch could be partly due to the fact that in figure 2d and e we analyzed proportions of specific CD4+ T cell subsets in blood, namely Th and Tfh, while in supplementary figure 4 we compared the capacity to secrete pro-inflammatory cytokines by total CD4+ T cells upon in vitro stimulation. Moreover, discrepancies may be related to exhaustion of these T cells as they showed higher expression of PD-1.
We had already discussed this possibility in the manuscript, Results section 3.4 (page 9):
“Despite the aforementioned disturbances in the distribution of T cell populations, the secretion of interleukin (IL)-2, interferon gamma (IFNg) and tumor necrosis factor alpha (TNFa) by CD4+ or CD8+ T cells after polyclonal in vitro stimulation was similar between PSP and NSP, and corticosteroid treatment did not have a clear effect on cytokine synthesis (Figure S4). The lack of increased cytokine production by activated T cells in PSP, together with their elevated expression of PD-1, raises the possibility of these memory CD4+ and CD8+ T cells being exhausted.”
Question 4:
Figures 2 and 3 are partially redundant. Authors should not show the same data in two different figures. This needs to be reorganized.
Answer:
Several results were shown in both figure 2 and 3 and following the reviewer´s suggestion, we have removed this duplicity by incorporating the data on the 4 month follow up into Figure 2 (pink bars). Figure 3 has been removed and the new Figure 2 is shown below:
Figure 2. Immunological alterations in PSP include dysregulated innate and adaptive populations together with higher expression of activation factors. Comparison between NSP, PSP, TTP and TTP+4M of (a) conventional dendritic cells (CD11b+ DC), natural killer T lymphocytes (NKT), plasmacytoid DC (CD123+ pDC) and innate lymphoid cells (ILC); (b) naïve, memory and follicular helper T CD4+ lymphocytes; (c) expression of HLA-DR and CD38 and expression of CD95 and PD-1 on CD4+ T lymphocytes; (d) subsets of memory CD4+ T cells based on CXCR3 and CCR6 expression; (e) subsets of follicular helper T CD4+ lymphocytes; (f) regulatory T cells (Treg); (g) naïve, central memory (TCM), effector memory (TEM) and effector memory re-expressing CD45RA (TEMRA) CD8+ T cells; and (h) naïve, non-class-switched memory (NCS), class-switched memory (CS) and plasmablast B cells. Grey regions depict percent populations ranges, if available, obtained from groups of representative healthy donors in our clinical immunology laboratory. In (b) tSNEs were performed by using all 37 fluorochromes (see Methods and extended data Figure 6); and in (c) tSNEs were performed in total CD4+ cells and using CXCR5, CD27, CCR7, CD45RA, CCR6 and CXCR3. *, p<0.05; **, p<0.01; ***, p<0.001.
We also included results at month 4 after corticosteroid treatment in supplementary figure 3:
Figure S3. Expression of activation and senescence markers on CD8 T cells. *, p<0.05; **, p<0.01; ***, p<0.001.
